# Frequency-dependent decoupling of domain-wall motion and lattice strain in bismuth ferrite

Lisha Liu[1], Tadej Rojac[2], Dragan Damjanovic [3], Marco Di Michiel[4] & John Daniels[1]

Dynamics of domain walls are among the main features that control strain mechanisms in ferroic materials. Here, we demonstrate that the domain-wall-controlled piezoelectric behaviour in multiferroic $BiFeO_3$ is distinct from that reported in classical ferroelectrics. In situ X-ray diffraction was used to separate the electric-field-induced lattice strain and strain due to displacements of non-180° domain walls in polycrystalline $BiFeO_3$ over a wide frequency range. These piezoelectric strain mechanisms have opposing trends as a function of frequency. The lattice strain increases with increasing frequency, showing negative piezoelectric phase angle (i.e., strain leads the electric field), an unusual feature so far demonstrated only in the total macroscopic piezoelectric response. Domain-wall motion exhibits the opposite behaviour, it decreases in magnitude with increasing frequency, showing more common positive piezoelectric phase angle (i.e., strain lags behind the electric field). Charge redistribution at conducting domain walls, oriented differently in different grain families, is demonstrated to be the cause.

[1] School of Materials Science and Engineering, UNSW, 2052 Sydney, Australia. [2] Electronic Ceramics Department, Jozef Stefan Institute, 1000 Ljubljana, Slovenia. [3] Group for Ferroelectrics and Functional Oxides, Swiss Federal Institute of Technology in Lausanne—EPFL, 1015 Lausanne, Switzerland. [4] ESRF—The European Synchrotron, 38043 Grenoble, France. Correspondence and requests for materials should be addressed to J.D. (email: j.daniels@unsw.edu.au)

Ferroic domain walls are naturally occurring nanoscale interfaces that can possess distinct properties from their parent materials[1–3]. Due to their length-scale, they have generated great interest for applications such as domain-wall nanoelectronics[2–6]. The ferroelectric material $BiFeO_3$ has been demonstrated to possess enhanced electrical conductivity at domain walls (relative to the inner domain away from the domain walls)[2]. This material also displays Maxwell–Wagner-like frequency dispersion in its macroscopic piezoelectric response[7]. This dispersion is different from a prototypical frequency dependent behaviour of the piezoelectric coefficient in classical ferroelectric materials, such as soft $Pb(Zr_xTi_{1-x})O_3$ (PZT). In PZT, the piezoelectric response exhibits a linear-logarithmic dependence, that is interpreted by field-induced motion of ferroelectric and ferroelastic domain walls in a medium with random pinning centres[8]. The Maxwell–Wagner-like frequency dispersion in bulk polycrystalline $BiFeO_3$ has been suggested to be originated from the conductive domain walls[7].

Maxwell–Wagner-like dispersion of the piezoelectric coefficient, similar to that shown for $BiFeO_3$, has been previously observed in polymer–polymer and polymer–ferroelectric composites[9–11] and several ferroelectrics with Aurivillius structures[12]. The Maxwell–Wagner effect[13], i.e., charge accumulation and its decay at interfaces between constituent components with different dielectric and electric conduction properties inside a material, can be easily understood for layered Aurivillius structures with strongly anisotropic conductivity and in heterogeneous systems such as composites[12]. However, in polycrystalline ferroelectrics with perovskite structures such as $BiFeO_3$, where significant anisotropy in the bulk conductivity is not expected, the reasons for Maxwell–Wagner effects are not so obvious. Moreover, the macroscopic Maxwell–Wagner-like piezoelectric dispersion in $BiFeO_3$[7] is unique from that previously observed in other materials. This reflects in the remarkable nonlinearity with respect to the driving field amplitude at low frequencies (<10 Hz) and the negative piezoelectric phase angle (phase leading) at weak fields (i.e., the piezoelectric strain response leads the driving electric field). Pivotal to future applications not only of $BiFeO_3$ but also other oxides exhibiting domain wall conduction is a thorough understanding of the origins of these behaviours and their relation to conductive domain walls.

In polycrystalline ferroelectric materials, the converse piezoelectric response to an applied electric field has its origin in several structural features, for example, small displacements of atoms under external fields in the crystal unit cell, i.e., the lattice strain, motion of non-180° domain walls resulting in a change in ferroelectric/ferroelastic domain texture, and electric-field-induced phase transformations. Methods for quantification of lattice and domain wall motion induced strains have been previously used to investigate the strain response of polycrystalline materials[14,15]. Unlike thin films[16], bulk $BiFeO_3$ does not undergo crystallographic phase transformations under fields approaching the breakdown field of the material[17] and thus strain due to lattice distortion and motion of non-180° domain walls are the majority contributors to its macroscopic piezoelectric response. These strain mechanisms in other ferroelectrics, as observed by in situ X-ray diffraction (XRD) field-dependent measurement on PZT and $PbTiO_3$–$BiScO_3$, are considered to be interdependent and coupled through intergranular elastic constraints between neighbouring grains or within clusters of grains[18–20]. Reports on the frequency dispersion of individual strain mechanisms are limited. Previous work on PZT[21] and $36\%BiScO_3$–$64\%PbTiO_3$[22] materials showed decreased domain wall motion with increasing frequency. However, lattice strain is either not provided or is independent of frequency.

Here we demonstrate by using time-resolved in situ XRD and analytical modelling that the distinct frequency dependent piezoelectric behaviour in polycrystalline $BiFeO_3$ is due to conducting domain walls. By experimentally separating the lattice strain from the change in non-180° domain texture over the frequency range from 0.01 to 1000 Hz, we show that the two strain mechanisms in different grain orientations are decoupled as a function of frequency, meaning that the amplitudes of these two strain mechanisms have opposite trends with respect to variation in driving frequency. Surprisingly, the lattice strain increases in magnitude with increasing frequency for grains with a $\{100\}_{pc}$ direction aligned with the electric field, showing unusual negative piezoelectric phase angle and thus a strain response that leads the external field. This is the first direct observation of a strain mechanism showing phase leading behaviour using the XRD method. In contrast, the strain coming from non-180° domain wall motion in $\{111\}_{pc}$ grains decreases with increasing frequency, showing a more common positive piezoelectric phase angle, meaning lagging of the strain response to the external electric field. In addition to experimental in situ XRD data, we present here an analytical model based on the domain wall conductivity to show the origin of the microscopic strain decoupling as a function of frequency and negative phase angle of the piezoelectric response of $BiFeO_3$. Charge redistribution at the domain walls, oriented differently with respect to the applied field vector in different grain families, causes complex time-dependent internal electric fields, effectively resulting in redistribution of these fields in different grains. The revealed mechanism may play an important role in ferroelectrics exhibiting significant local variations in electrical conductivity, particularly those characterized by enhanced conduction at domain walls, and thus offers a new approach based on conducting domain walls for influencing the electromechanical properties of ferroelectrics.

## Results

**In situ time-resolved XRD.** Figure 1a displays a representative diffraction pattern integrated from the diffraction image obtained during in situ sub-coercive electric-field cycling on poled $BiFeO_3$ samples (see Methods). Figure 1b–d shows the profile fitting of selected diffraction peaks using Gaussian functions. No impurity phase was found within the detection limit of the instrument, confirming the high phase purity of the material. The fitted parameters including position of individual peaks ($\theta_{hkl}$) and $111_{pc}/11\bar{1}_{pc}$ peak intensities, which vary upon field application, were utilized to obtain lattice strains, $\varepsilon_{hkl}$, and non-180° domain texture, $f_{111}$, during the electric-field cycling using following equations:

$$\varepsilon_{hkl} = -\left(\theta_{hkl}^E - \theta_{hkl}^0\right)\cdot\cot\theta_{hkl}^0 \qquad (1)$$

$$f_{111} = 4\frac{\frac{I_{111}^E}{I_{111}^0}}{\frac{I_{111}^E}{I_{111}^0} + 3\cdot\frac{I_{111}^E}{I_{11\bar{1}}^0}} \qquad (2)$$

where the superscript indicates the state of external electric field application (either zero or non-zero field $E$) and subscript denotes the crystallographic $hkl$ plane. Equation (2) is a multiple of a random distribution (MRD) method used to quantify domain texture for rhombohedral structures[15]. In contrast to $f_{111} = 1$ MRD for as-processed unpoled ceramic, the $f_{111}$ for the initial poled state of the measured sample was 2.5 MRD.

Figure 2a–c shows the driving field, $200_{pc}$ peak position, and $111_{pc}/11\bar{1}_{pc}$ peak intensities during the 1 Hz electric field cycle, respectively. The calculated lattice strain from peak positions and $f_{111}$ from the interchange of intensities, using equations (1) and

(2), are presented in Fig. 2d and e, respectively. These data are then fitted with a sinusoidal curve to extract the amplitude variation of the electric-field-induced lattice strains, $\varepsilon_{hkl}$, and changes of the $111_{pc}$ domain texture, $\Delta f_{111}$, at this specific frequency (the same procedure is used for all frequencies). Changes of lattice strains, $\varepsilon_{hkl}$, from each individual peak (see Supplementary Figure 1), were then combined using a weighted average method to obtain the total lattice strain contribution,

$\varepsilon_{intrinsic}$, in the field direction of the sample by the following equation (3)[14]:

$$\varepsilon_{intrinsic} = \sum_{hkl} f_{hkl}(0) m_{hkl} \varepsilon_{hkl} / \sum_{hkl} f_{hkl}(0) m_{hkl} \qquad (3)$$

where $f_{hkl}(0)$ is the domain texture along the field direction (i.e., when the angle between applied field vector, **E**, and scattering vector, **q**, is 0°) and can be measured using diffraction techniques[15,23]. The multiplicity factor of ($hkl$) reflection, $m_{hkl}$, is the corresponding total number of identically spaced planes.

Figure 3a–c shows in situ total lattice strain, $\varepsilon_{intrinsic}$, change in non-180° domain texture, $\Delta f_{111}$, macroscopic strain, $\varepsilon_{macro}$, as measured from sample surface displacement, and tangent of the piezoelectric phase angle, tanδ, as a function of frequency obtained during 6 kV mm$^{-1}$ unipolar electric-field cycling. The macroscopic strain is the field response of the whole bulk sample and thus represents a complex convolution of all strain mechanisms, including lattice strain and local domain-wall induced strain from individual grains. Distinct frequency dispersions of the strains are observed in the measured frequency range (Fig. 3a, b).

From Fig. 3a, it can be seen that lattice strain (magenta circles) and change in non-180° domain texture (blue diamonds) are decoupled as a function of frequency: they show different trends in their fractional contributions to the macroscopic strain as a function of frequency. The total lattice strain increases with increasing frequency of the field, while the change in non-180° domain texture shows a decrease in magnitude with increasing frequency. A clear frequency-dependent decoupling of these strain mechanisms occurs at frequencies below ~10 Hz. For

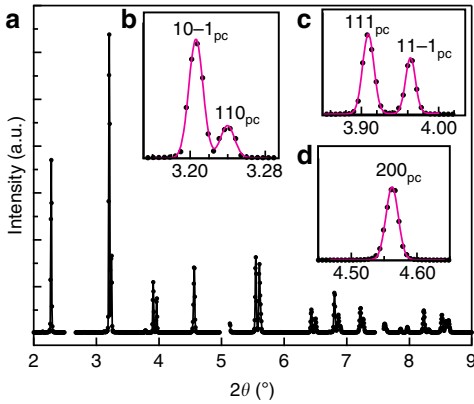

**Fig. 1** X-ray diffraction results. **a** Integrated segment of diffraction images of BiFeO$_3$ driven at 6 kV mm$^{-1}$ and 1 Hz cyclic electric field (see Methods). Insets indicate profile fitting using Gaussian peak functions on **b** $110_{pc}$, **c** $111_{pc}$, and **d** $200_{pc}$, enabling the extraction of peak position, $2\theta$, and intensities of these individual single and double peaks

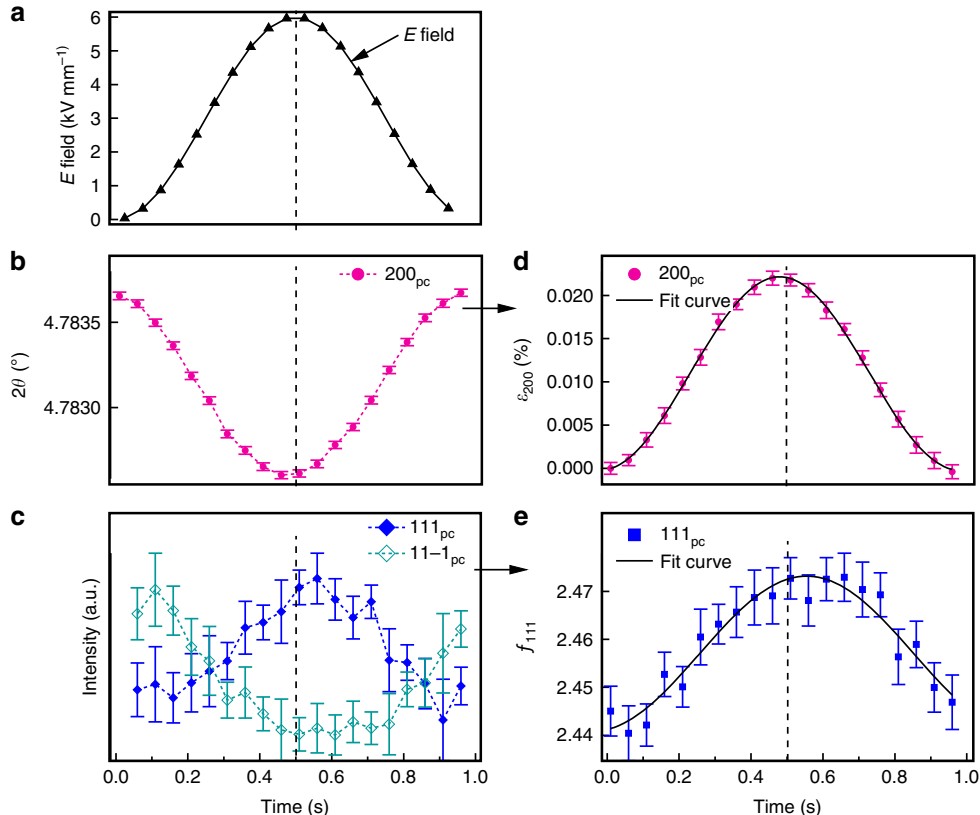

**Fig. 2** Strain response to sinusoidal electric field. **a** Sinusoidal electric field, $E$, demonstrated for the 1 Hz measurement; **b** $200_{pc}$ peak position, $2\theta$; **c** intensities of $111_{pc}/11\bar{1}_{pc}$ reflections, and calculated **d** $200_{pc}$ lattice strain, $\varepsilon_{200}$; and **e** non-180° $111_{pc}$ domain texture, $f_{111}$. The error bars arise from profile fitting using Gaussian peak functions on individual single and double peaks. The dashed line is used here to indicate amplitude of electric field at ~0.5 s for the 1 Hz electric field

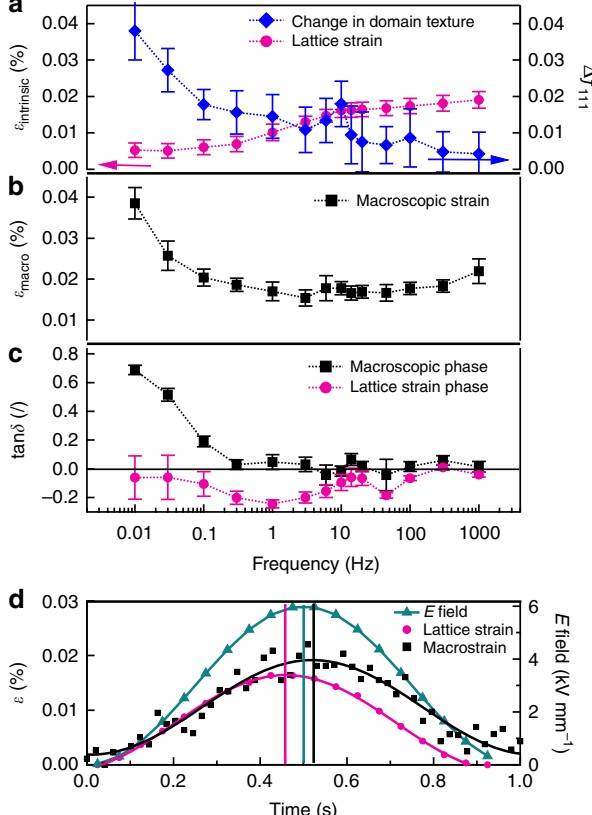

**Fig. 3** Effect of electric field cycling as a function of frequency. **a** Calculated total lattice strain (magenta circles), $\varepsilon_{intrinsic}$, using equation (3), and change in non-180° $111_{pc}$ domain texture (blue diamonds), $\Delta f_{111}$, using equation (2); **b** measured in situ macroscopic strain, $\varepsilon_{macro}$, by an optical displacement sensor coupled to the sample surface during XRD experiments (see Methods); **c** tangent of the piezoelectric phase angle, $\tan\delta$, of macroscopic strain (black squares) and lattice strain (magenta circles) during application of 6 kV mm$^{-1}$ unipolar sinusoidal electric field, $E$; and **d** time dependence of driving electric field (turquoise triangles), macroscopic strain (black squares), and lattice strain (magenta circles) responses demonstrated at 1 Hz, showing lagging (black straight line) and leading (magenta straight line) between strain and sinusoidal field signals. The errors of the lattice strain, change of non-180° $111_{pc}$ domain texture, macroscopic strain, and phase angle arise from sinusoidal curve fitting on these responses during application of driving cyclic field

frequencies above 10 Hz, the trends with respect to increasing frequency between the two microscopic strain mechanisms are still opposite but strain changes are smaller.

As shown in Fig. 3b, the macroscopic strain decreases from 0.039 to 0.015% with increasing frequency from 0.01 to ~3 Hz. Within experimental error (see Methods), the strain then stabilizes at 0.017% for frequencies above ~6 Hz. The frequency below which the lattice strain, change in domain texture and macrostrain becomes dispersive, i.e., 10 Hz, is consistent with previously reported ex situ macroscopic measurements of the direct and converse piezoelectric response of BiFeO$_3$[7].

As expected for the field used in this experiment (6 kV mm$^{-1}$, 0.75 E$_c$)[7], Fig. 3c (black squares) shows that the phase angle of the macroscopic strain is positive (meaning that the strain lags behind the driving field) and increases as the frequency is decreased below ~0.3 Hz. For all other frequencies measured (>0.3 Hz), $\tan\delta$ is approximately zero within experimental error. In contrast, the phase angle of the lattice strain is negative for the majority of the frequency range (meaning that strain leads the

driving field), with its minimum value at approximately 1 Hz (Fig. 3c, magenta circles). It is worth noting that the negative phase angle was also observed in the direct $d_{33}$ measurements at small unipolar stresses[24] and in the converse $d_{33}$ with bipolar electric fields when the response is extrapolated to zero field[7]. At low stresses and electric fields, the negative phase of both the direct and converse macroscopic strain can be related to the observed phase leading of lattice strain directly measured by in situ XRD here. Increasing driving stresses and fields will enhance the domain wall displacements, which results in positive phase angle that then dominates the direct and converse macroscopic responses. All of these different and mutually consistent experiments confirm the presence of negative phase angle in the piezoelectric response of BiFeO$_3$. The reproducibility of the negative phase angle of the lattice strain over multiple cycles at frequencies below 1 Hz and over all orientations with respect to the electric field vector **E** (i.e., the phase angle of lattice strain at all angles between the $200_{pc}$ diffraction vector **q** and **E**) at both low and high frequencies (e.g., 1 and 100 Hz) are demonstrated in Supplementary Table 1 and Supplementary Figure 2, respectively.

The phase lagging and phase leading of the macroscopic strain and microscopic lattice strain with respect to the driving field, respectively, is demonstrated for the 1 Hz measurement in Fig. 3d. This unusual behaviour (negative phase angle) is physical for the piezoelectric response[25–27] and it has been observed experimentally in systems where conductivities with different time constants occur in different regions of a material[12]. In those cases, the negative phase angle is a manifestation of the varying electric field distribution in the system during the electric field cycle.

**Maxwell–Wagner analytical model and Rayleigh relationship calculations.** Charge accumulation and redistribution through local conductive paths in the material can affect the temporal dependence of internal electric fields, leading to the so-called Maxwell–Wagner dielectric[28] and piezoelectric relaxation[12,29]. Such dispersive behaviour was previously observed in the macroscopic dielectric[30] and piezoelectric response of BiFeO$_3$ ceramics[7]. To confirm consistency with these previous results, the dielectric permittivity and piezoelectric coefficient were measured on a sample from the same batch as that analysed by the in situ XRD experiments under the same condition (unipolar 6 kV mm$^{-1}$ field). The permittivity was also measured at weaker field amplitude (bipolar 0.02 kV mm$^{-1}$). All these results clearly and independently confirm a consistent Maxwell–Wagner-like frequency dispersion in the dielectric and piezoelectric responses (see Supplementary Figures 3–5 and detailed discussion thereof). Considering the above discussion, local regions within the polycrystalline matrix that have dispersive mechanisms with different time constants are likely present in BiFeO$_3$. While this is unexpected in a homogeneous phase-pure material with simple perovskite structure where significant anisotropy in the conductivity is not present, the anisotropy in conductivity, as we show next, may be caused by the presence of conducting domain walls.

In BiFeO$_3$, enhanced domain wall conductivity with respect to domain region has been observed in both thin films[2] and ceramics[31]. Defects are assumed responsible for the domain wall conductivity by preferentially concentrating in the domain wall region[31]. These defects in BiFeO$_3$ ceramic sintered in O$_2$ or air have been identified as bismuth vacancies and Fe$^{4+}$ cations (representing oxidized states of Fe$^{3+}$ cations). This allows p-type hopping conduction at domain walls due to electron holes associated with Fe$^{4+}$ [31]. For the same batch of poled material analysed by in situ XRD experiments, the enhanced electrical conductivity at domain walls was confirmed by combined

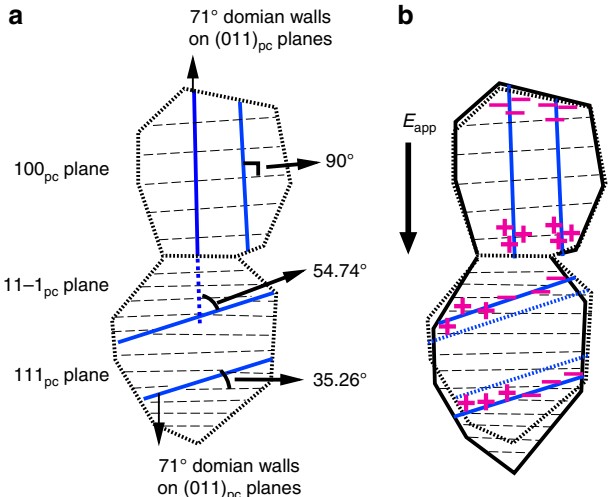

**Fig. 4** Schematic of two representative grains in poled BiFeO3. **a** The diffraction planes and orientations of 71° domain walls, occurring on $(011)_{pc}$ planes, in $\{100\}_{pc}$ and $\{111\}_{pc}$ grains. In $\{100\}_{pc}$ grain (top grain), the 71° domain wall (solid blue lines) is perpendicular to the $100_{pc}$ diffracting planes (black dashed line). In the $\{111\}_{pc}$ grain, the 71° domain walls separate $111_{pc}$ and $11\bar{1}_{pc}$ diffracting planes and form an angle of 35.26°. The angle between 71° domain walls in $\{100\}_{pc}$ and $\{111\}_{pc}$ grains is 54.74°; **b** charge distribution on conductive domain walls under applied external electric field, $E_{app} = E_0\sin(\omega t)$. Charge redistribution rate is different for grains with different crystallographic orientations, i.e., $\{100\}_{pc}$ and $\{111\}_{pc}$. Their dielectric permittivities, electrical conductivities and piezoelectric coefficients are thus different, represented by $\kappa_{100}$ and $\kappa_{111}$, $\sigma_{100}$ and $\sigma_{111}$, and $d_{100}$ and $d_{111}$, respectively. This will result in different effective fields in individual grains, represented by $E_{100}$ and $E_{111}$. The grain elongations upon field application, indicated by the solid black shapes, are due to the piezoelectric effect in the $\{100\}_{pc}$ grain and non-180° domain wall motion in the $\{111\}_{pc}$ grain

conductive atomic force microscopy (c-AFM) and piezo-response force microscopy (PFM) (see Supplementary Figure 6). We thus propose that the conductivity at domain walls plays a key role in the observed frequency-dependent strain decoupling, as explained in the model presented here, though other origins of local conductivity, such as that at grain boundaries and/or pores, may also contribute to the effect to some extent[7].

Figure 4a, b shows two representative grains that have different crystallographic orientations in rhombohedral polycrystalline BiFeO3, i.e., $\{100\}_{pc}$ and $\{111\}_{pc}$, where the directions represent the crystallographic planes aligned perpendicular to the applied electric field vector. In this figure, the diffracting planes are represented by black dashed lines and the domain walls by blue solid lines. In rhombohedral BiFeO3, 71° domain walls occur on $\{101\}_{pc}$ planes and 109° on $\{100\}_{pc}$ planes[32]. This means that the conductive domain walls of the two representative grain families occur at different angles to the applied field. As an example, Fig. 5 shows one possible orientation of 71° conductive domain walls occurring on $(011)_{pc}$ planes. These domain walls run parallel to the applied external electric field vector in the $\{100\}_{pc}$ oriented grain family (see Fig. 4 top grain). However, in the $\{111\}_{pc}$ oriented grain family, $(011)_{pc}$ planes have an angle of 35.26° with $(111)_{pc}$ planes (see Fig. 4 bottom grain), so the angle difference of the conductive path to the applied external electric field vector is 54.74° (Fig. 4a). This difference in the orientation of the conducting walls between various grain orientations of the material, with respect to the external field vector, is what is proposed here to cause a difference in the conductivity between different grains. The difference in the conductivity is responsible

for the redistribution of the effective field in each grain, producing Maxwell–Wagner-like frequency dispersion in an otherwise homogeneous material like BiFeO3.

When an external field is applied, mobile charges in the domain wall region can redistribute via hole hopping. In the case of the $\{100\}_{pc}$ grain family, where the conducting walls are parallel to the applied field, the driving force for this redistribution is maximized. However, in the $\{111\}_{pc}$ grain family the domain walls are not parallel to the external field and the driving force is reduced. Thus, the conductivities of two grain families are different. This process is confirmed by the model which is shown next.

The redistribution of electric fields in these grains is analogous to effective fields in circuits where leaky capacitors (i.e., a capacitor and a resistor in parallel) with different capacitance and resistivity are connected in series. We largely simplify the case to a two-grain system as shown in Fig. 4, without considering all other grain orientations, different domain clusters within grains, intergranular elastic coupling, the transverse piezoelectric response and elastic compliance of the grains (see Methods). In series connection, the external field over the two grains $E_{app}$ is the weighted sum of the field in the individual grains $E_{100}$ and $E_{111}$ ($E_{app} = v_{100}E_{100} + v_{111}E_{111}$, where $v_{100}$ and $v_{111}$ is the volume fraction of each grain), while the charge density at the surface of different grains is the same ($\kappa_{100}E_{100} = \kappa_{111}E_{111}$), where $\kappa_{hkl}$ is dielectric permittivity for each grain. Assuming for simplicity that the only electrical loss mechanism is the conductivity, the complex permittivity can be expressed as $\kappa_{hkl} = \kappa'_{hkl} - i\sigma_{hkl}/\omega$, where $\sigma_{hkl}$ is the conductivity in the grain with $hkl$ orientation and $\omega$ is the angular frequency of the driving field. These expressions can be used to calculate the effective electric fields for each grain as a function of the frequency and external field, $E_{app}$ (see Methods).

The effective electric fields can then be used to calculate the frequency dependent strain, $\varepsilon_{hkl}$, in each grain using piezoelectric equations[33]:

$$\varepsilon_{100} = d_{100}E_{100} = d_{100}\left(E'_{100} - iE''_{100}\right) = \varepsilon'_{100} - i\varepsilon''_{100} \quad (4)$$

$$\varepsilon_{111} = d_{111}E_{111} = d_{111}\left(E'_{111} - iE''_{111}\right) = \varepsilon'_{111} - i\varepsilon''_{111} \quad (5)$$

The subscripts, $hkl$, represent the grain orientation with respect to the external field vector. $E'$ and $E''$ are the real and imaginary components of the electric field, respectively, $d_{100}$ and $d_{111}$ are the piezoelectric coefficients incorporating the dominant strain mechanisms for $\{100\}_{pc}$ and $\{111\}_{pc}$ oriented grains. Due to the orientation of the grain, $d_{100}$ is inhibited from having a domain wall motion component, as the spontaneous strain resolved along the field direction is always the same for all domain orientations. Meanwhile, $d_{111}$ is likely to be dominated by the domain wall motion component and its field and frequency dependence is not considered in equation (5), but is incorporated in a later stage (see equation (8)). The resulting strain is also complex, consisting of a real ($\varepsilon'$) and an imaginary component ($\varepsilon''$).

The phase angles of the strains in different grains can be obtained as:

$$\tan\delta_{\varepsilon_{100}} = \varepsilon''_{100}/\varepsilon'_{100} = E''_{100}/E'_{100} \quad (6)$$

$$\tan\delta_{\varepsilon_{111}} = \varepsilon''_{111}/\varepsilon'_{111} = E''_{111}/E'_{111} \quad (7)$$

Adopting an external electric field $E_{app}$ of 6 kV mm$^{-1}$, $d_{100}$ and $d_{111}$ of 45 and 30 pm V$^{-1}$, relative permittivity values of $\kappa_{100} = 40$ and $\kappa_{111} = 30$ and relative conductivity values (with respect to the

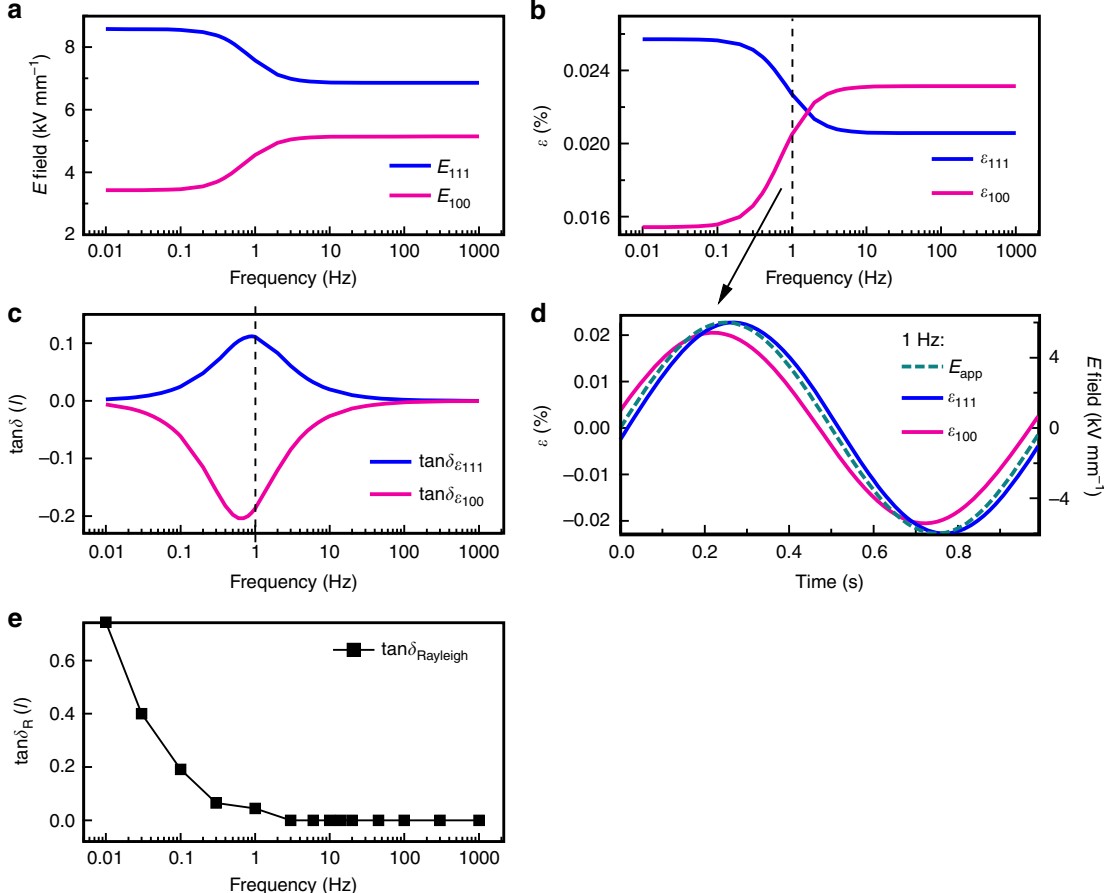

**Fig. 5** Maxwell–Wagner analytical model and Rayleigh relationship calculations. **a** Redistributed effective fields, $E_{100}$ and $E_{111}$, in the two representative grains due to domain wall conductivity; **b** frequency dispersion and frequency-dependent decoupling of strain responses, $\varepsilon_{100}$ and $\varepsilon_{111}$, in $\{100\}_{pc}$ and $\{111\}_{pc}$ grain families calculated from effective field redistribution using piezoelectric equations; **c** phase angle, $\tan\delta_{\varepsilon_{100}}$ and $\tan\delta_{\varepsilon_{111}}$, of the strains in each grain family; and **d** time-dependent electric field and strains in two grains demonstrated for 1 Hz driving frequency. Calculations were made using equations (4)–(7) with relative permittivity $\kappa_{100} = 40$, $\kappa_{111} = 30$, relative conductivity $\sigma_{100} = 250$, $\sigma_{111} = 100$, piezoelectric coefficient $d_{100} = 45$ pm V$^{-1}$, $d_{111} = 30$ pm V$^{-1}$, and volume fractions of each grain $v_{100} = v_{111} = 0.5$. **e** Rayleigh phase angle due to the significant non-linearity at frequencies below ~1 Hz calculated from the experimental irreversible parameter, $\alpha$ (see Supplementary Figure 8)

vaccum permittivity) $\sigma_{100} = 250$ and $\sigma_{111} = 100$ (see Supplementary Figure 7 and Methods), results of the calculations for different grains from this analytical model are shown in Fig. 5. Figure 5a shows the calculated effective electric fields (see Methods), while Fig. 5b shows the strain response of the two grain orientations over the frequency range of interest using equations (4) and (5). The analytical model reproduces the frequency-dependent decoupling of strains in the two grain families (Fig. 5b; $100_{pc}$ and $111_{pc}$ strains have opposite trends with decreasing frequency below ~10 Hz), caused by the effective field redistribution (Fig. 5a). The corresponding phase angle of strain response in each grain can be obtained by equations (6) and (7) as given in Fig. 5c. The phase angle shows a negative peak for the $\{100\}_{pc}$ grain and a positive peak for the $\{111\}_{pc}$ grain (Fig. 5c). This highlights the leading and lagging of individual strain mechanisms relative to the external electric field. One example of this leading and lagging during an individual electric field cycle is calculated for 1 Hz driving field in Fig. 5d, comparable to the experimental data shown in Figs. 2d, e and 3d.

At low frequencies, the experimental data, Fig. 3b, c, show a large increase in macroscopic strain magnitude and positive phase angle that is not fully accounted for in the Maxwell–Wagner type analytical model. This discrepancy is explained by the frequency dependence of the non-linearity in the piezoelectric response

shown in Supplementary Figure 8. Below 0.3 Hz, the non-linearity is significant, increasing with decreasing frequency, while at frequencies above 1 Hz the non-linearity is suppressed. This explains the increase of macroscopic strain at low frequencies (Fig. 3b, <1 Hz). The reason for this behaviour is that domain wall conductivity which we have shown to be active below 1 Hz allows depinning of domain walls, increasing their mobility as shown in ref. [7]. This enhanced domain wall motion at low frequencies causes an increasing phase angle as estimated using equation[34]:

$$\tan\delta_R \approx \delta_R = \frac{4\alpha(f)E_{max}}{3\pi d_{33}} \qquad (8)$$

This equation is derived from Rayleigh relationships that describe non-linear and hysteretic movement of domain walls, where $\alpha$ is the frequency-dependent Rayleigh irreversible parameter, $E_{max}$ is the applied field amplitude and $d_{33}$ is the macroscopic piezoelectric coefficient. The calculated Rayleigh phase angle from experimental $\alpha$ (see Supplementary Figure 8) is shown in Fig. 5e. This calculation follows closely to the observed increase in phase angle of the macroscopic strain shown in Fig. 3c.

The Maxwell–Wagner analytical model and Rayleigh relationship applied at low frequencies reproduce the key features of the

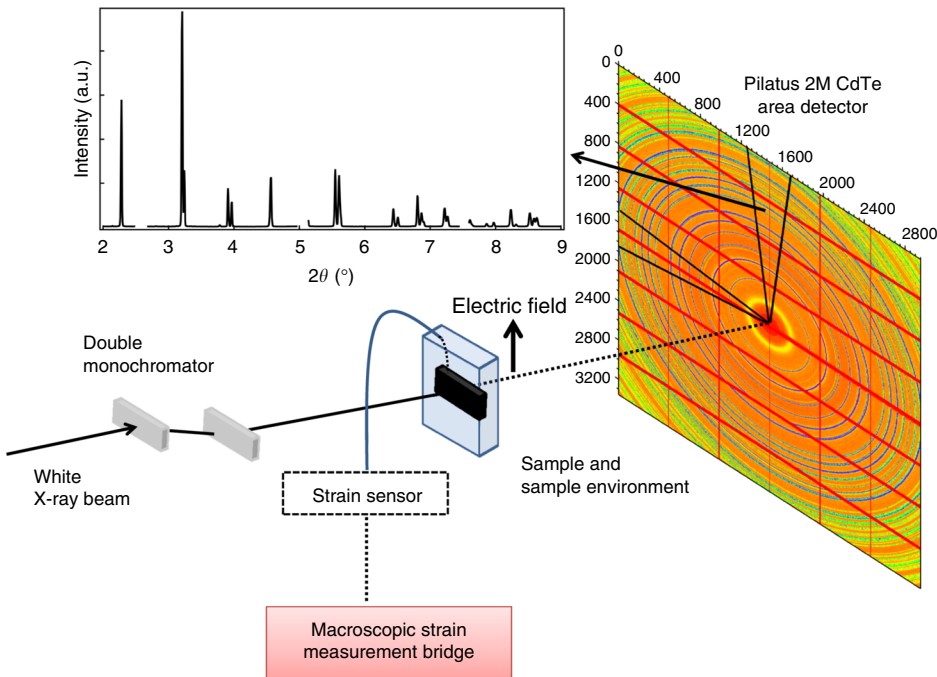

**Fig. 6** Schematic of the in situ XRD experimental setup. Samples were placed in the sample chamber and were bathed in silicon oil. The 2D image is obtained by a Dectris Pilatus3 X CdTe detector (see Methods) and the diffraction pattern is radially integrated from a wedge of the 2D diffraction image

experimental data over the full frequency range measured, i.e., decoupling of strain mechanisms with frequency, and phase leading of the lattice strain. The frequency-dependent decoupling of strains shown here is counter-intuitive to the usually assumed interdependence and continuous grain interactions, that cause coupling of the strain mechanisms by intergranular stress between neighbouring grains[18,19]. More specifically, the frequency dispersion of lattice strain (decrease of lattice strain with decreasing frequency) is distinct from previous reports (frequency-independent lattice strain)[22]. Based on the above analytical model, this difference is explained by domain walls in $BiFeO_3$ acting as conductive paths. Therefore, shielding of the applied external electric field occurs in given grain families, especially at lower frequencies with more pronounced domain wall motion and charge migration, resulting in the observed decrease of lattice strain with decreasing frequency. Moreover, this phenomenon is accompanied by the phase of the lattice strain leading the external electric field.

## Discussions

Electric-field-induced microscopic strain mechanisms and the macroscopic strain of polycrystalline $BiFeO_3$ have been studied as a function of frequency at sub-coercive field values to provide insight to the origin of Maxwell–Wagner-like dispersion in its converse piezoelectric response. Two features of the strain response of $BiFeO_3$ are observed, i.e., the frequency-dependent decoupling of strain mechanisms (lattice strain and local domain texture) in different grain families and the negative phase angle (phase leading) of the electric-field-induced lattice strain of $\{100\}_{pc}$ oriented grains. The latter is the first XRD observation of strain response temporally leading the external stimulus. The reproducibility of these unusual behaviours was confirmed by repeating measurements at 20 Hz on single samples, and over the full frequency range on multiple samples (Supplementary Table 2, Supplementary Figures 9, 10). These measurements show that grain interactions in the $BiFeO_3$ ceramic are stable during sub-coercive electric-field cycling. Breaking of grain-scale mechanical coupling, for example

by cracking, would be expected to cause large changes in response when measurements are repeated at the same frequencies on single samples and is thus excluded here as a reason for frequency-dependent decoupling of microscopic strain mechanisms.

Analytical modelling and the totality of the experimental data show that the Maxwell–Wagner relaxation in the converse piezoelectric response can be due to conductive domain walls in $BiFeO_3$ ceramic. Local variations in conductivity, related to orientation of domain walls with respect to the external field in grains, modify the internal field distribution in different grain orientation families. Given the hypothesis that domain wall conductivity is the main reason for the Maxwell–Wagner-like dispersion, it is foreseeable that the frequency dispersion of piezoelectric properties in $BiFeO_3$ can be highly sensitive to processing conditions that impact both defect type and concentration as well as microstructural features. The sensitivity of defect formation when the material is processed under different $O_2$ partial pressures and sintering temperatures is known to directly influence the domain wall conductivity in $BiFeO_3$ ceramics and thin films[31,35]. In addition, grain boundaries and pore surfaces, may show conductive properties. However, contributions from these regions are expected to be isotropic and thus they cannot explain the difference in the response magnitude observed in different grain orientation families. Different processing conditions or sintering temperatures would also result in variations in grain size and domain structures, which according to our model, are also expected to affect the frequency dispersion of the piezoelectric response. In our case, to test the reproducibility of XRD measurements and the observed increase of lattice strain with increasing frequency, samples were sintered in a narrow temperature window (between 745 and 780 °C), resulting in similar grain sizes and domain configurations (see Supplementary Figures 11, 12). Accordingly, the frequency-dependent behaviour of these samples is qualitatively similar, showing good reproducibility of the in situ results (Supplementary Figure 13).

The present findings help further understanding of grain-scale mechanics of polycrystalline piezoelectric materials. It shows that

control of domain wall conductivity, for example by adjusting processing atmospheres, may tune the properties of bulk BiFeO₃. This plays a role in influencing its macroscopic properties in a far more diverse and important manner than so far considered. Domain wall conductivity has been observed in numerous ferroelectric materials in addition to BiFeO₃[2,35], including BaTiO₃ crystals[36], KTiOPO₄ crystals[37], YMnO₃ single crystals[38,39], ErMnO₃ crystals[40], LiNbO₃ single crystals[41], and Pb(Zr₀.₂Ti₀.₈)O₃ thin films[42]. It would be interesting to see if those materials can also show the same frequency-dependent decoupling between the lattice and non-180° domain wall motion generated strain as in BiFeO₃. Potential novel applications of conducting domain walls in nanoelectronics have been discussed[3] and proposed[4] in the literature. Our findings add a new dimension to nanoengineering conductive domain walls, and thus further the development of domain-wall-based nanoelectronics in addition to providing a novel control mechanism for the frequency dependent properties of bulk materials.

## Methods

**In situ experiments**. High-purity bulk BiFeO₃ ceramics were prepared by the solid state method utilizing reactive sintering of a mixture of $Bi_2O_3$ (Alfa Aesar, 99.999%) and $Fe_2O_3$ (Alfa Aesar, 99.998%) raw powders. A small amount of $Co_3O_4$ (99%, Alfa Aesar), corresponding to 0.1 wt% of Co, was added to the mixture to reduce the electrical conductivity. The powder was then pressed to pellets uniaxially at 150 MPa. The obtained green pellet was then reactively sintered using tube furnace under different temperatures (745, 760 and 780 °C) to rule out sample variability during in situ XRD experiments.

The as-sintered ceramics were prepared into disks of 8 mm diameter and 0.5 mm thickness and electroded before being poled using a 12 kV mm⁻¹ electric field for 30 min. The measured $d_{33}$ values after poling of the samples were 42 ± 2 pC N⁻¹. Poled BiFeO₃ ceramics were then cut to dimensions of 1 mm × mm × 0.5 mm suitable for the in situ XRD measurements. The $d_{33}$ of the samples was checked after cutting to ensure no depoling occurred. Sub-coercive electric field cycling experiments were performed with a unipolar sinusoidal field of amplitude 6 kV mm⁻¹ (0.75 $E_c$) in the poling direction at the following frequencies: 1000, 300, 100, 45, 20, 14, 10, 6, 3, 1, 0.3, 0.1, 0.03, and 0.01 Hz. To obtain the required diffraction statistics for different frequencies, the measured number of cycles for the above frequencies were 2000, 600, 200, 90, 20, 20, 20, 10, 10, 10, 7, 3, 1 and 1, respectively. In general, frequencies where macroscopic strain data was measured for a larger number of cycles have smaller errors, as the fitting of the amplitude and phase could be performed more accurately. After the full cycling sequence, 20 Hz was selected to confirm the repeatability of the measured data and exclude radiation damage effects caused by the X-ray beam. Three batches (in total 17 samples) sintered under different temperatures (745, 760 and 780 °C) were used to carry out the in situ experiments.

High-energy XRD experiments were carried out at beamline *ID15A* of The European Synchrotron Radiation Facility. A schematic of the experimental setup for ID15A is shown in Fig. 6. Two X-ray beam energies were used for the measurements, 78.5 keV (0.15794 Å) and 75 keV (0.16531 Å). A beam size of approximately 200 μm × 200 μm was used.

The samples were mounted in a specifically designed chamber that allows it to be bathed in silicon oil during data collection while the electric field is applied[43]. The diffraction images were collected in the forward direction (transmission geometry) using a Dectris Pilatus3 X CdTe detector. Cyclic electric fields of different frequencies were generated by a function generator (Fluke PM5136) and amplified using a Trek 10/10 high voltage amplifier. This applied field was hardware synchronised with the framing of the detector diffraction images, such that diffraction information was obtained at specific time points within the electric field cycle. Detector parameters, including sample to detector distances, beam centre and detector tilts, were calibrated using standard cerium dioxide (NIST standard CeO₂) in *Fit2D*[44]. In this scattering geometry, each diffraction image contains full orientation dependent data of the scattering vector, **q**, angle with respect to applied electric field vector, **E**. Segments of the measured images were then integrated into sequential one-dimensional diffraction patterns using the calibrated detector information. Peak fitting was done sequentially for further interpretation in Igor Pro 7.0. Errors arising from the fitting of diffraction peaks were propagated through subsequent calculations.

Simultaneously, in situ macroscopic strain was measured using an optical displacement sensor coupled to the top surfaces of the sample. The displacement of the sample surface was used to calculate the macroscopic strain.

**Maxwell–Wagner analytical model and its parameters**. The Maxwell–Wagner analytical model is a simplification of the real case. To ease the otherwise complex interpretation of field redistribution and microscopic strains decoupling in the polycrystalline matrix of BiFeO₃, we did not take into account Rayleigh-like behaviour related to domain wall displacements, mechanical boundary conditions

and elastic anisotropy, or effects arising from the transverse piezoelectric response. The Rayleigh model is used, however, to interpret the large nonlinear increase in piezoelectric response at low frequencies. Electro-mechanical coupling among the grains in polycrystalline materials can be in principle added for a more complex and realistic model, for example similar to the one that has been reported by Turik et al.[29]. Other grain orientations, distributions of conductivities, fractional dynamics[45] or empirical Havriliak Negami equations could be considered for the complex permittivity of this system. However, the simplified model presented here captures qualitatively very well all essential features of the experimentally observed macroscopic behaviour.

The parameters involved in the analytical model include the dielectric permittivity $\kappa_{hkl}$, the electric conductivity $\sigma_{hkl}$, the piezoelectric coefficients $d_{hkl}$, and volume fraction $v_{hkl}$ of two grains with {111}ₚc and {100}ₚc orientation with respect to the external field, where $hkl$ represents the grain orientation. These values should reflect realistic physical values for BiFeO₃. For {111}ₚc grain, we use the intrinsic GHz relative dielectric permittivity of BiFeO₃, which is ~30 as reported previously[32]. For {100}ₚc grains, a higher relative permittivity is used, in this case a value of 40 is adopted for the model, considering that the transverse permittivity of a number of rhombohedral ferroelectrics with a 3 m symmetry (isostructural with BiFeO₃) is higher than the longitudinal permittivity[46]. The as-reported bulk electrical conductivity of BiFeO₃ ceramics and single crystals at room temperature are spread over several orders of magnitude, typically between ~10⁻² and ~10⁻¹⁰ Ω⁻¹ m⁻¹ [32,47–49]. The conductivity in the range of 10⁻⁹–10⁻¹⁰ Ω⁻¹ m⁻¹ was used in the calculations, consistent with the measured bulk electrical conductivity of our BiFeO₃ sample as shown in Supplementary Figure 7, and can reproduce the main features of the microscopic strains (the data shown in the main paper are relative conductivity of $\sigma_{111} = 100$ and $\sigma_{100} = 250$ for {111}ₚc grain and {100}ₚc grain, respectively). The piezoelectric coefficient of 30 pm V⁻¹ is used for {111}ₚc grain, and 45 pm V⁻¹ is used for {100}ₚc grain. A higher $d_{33}$ value is used for the {100}ₚc grain relative to the {111}ₚc grain for the same reasons as explained for the relative dielectric permittivity.

**Redistributed electric fields in different grain families**. The effective electric fields in different grain families, when we consider the ferroelectric material like BiFeO₃ exposed to an external electric field, is analogue to field redistribution in leaky capacitors with different capacitance ($C_i$) and resistivity ($R_i$) connected in series. The applied external voltage is $U_{app}$ and the field is $E_{app}$. $U_i$ and $E_i$ are the voltage and effective field on each capacitor. One further takes into account the fraction of different types of capacitance, $v_i$ (corresponding to the fraction volume of each grain family), which is proportional to their thickness $l_i$ if the effective area of all capacitors is the same. For series connection, the charge density ($D_i$) on each capacitor is equal, and the external voltage is the sum of the voltages on all capacitors. One obtains the following equations:

$$D_1 = \kappa_1 E_1 = \kappa_2 E_2 = D_2 \tag{9}$$

$$U_1 + U_2 = E_1 l_1 + E_2 l_2 = E_{app}(l_1 + l_2) = U_{app} \tag{10}$$

$$C_i = \varepsilon_i \frac{A}{l_i} \tag{11}$$

$$v_1 = \frac{l_1}{l_1 + l_2} \tag{12}$$

$$v_2 = \frac{l_2}{l_1 + l_2} \tag{13}$$

where $\kappa_i$ is dielectric permittivity of the capacitors. The electric fields on individual capacitors are thus:

$$E_1 = \frac{U_1}{l_1} = \frac{\kappa_2}{v_2 \kappa_1 + v_1 \kappa_2} E_{app} \tag{14}$$

$$E_2 = \frac{U_2}{l_2} = \frac{\kappa_1}{v_2 \kappa_1 + v_1 \kappa_2} E_{app} \tag{15}$$

Considering alternating external electric field with angular frequency $\omega$, and adding complex permittivity of the weakly conducting material as:

$$\kappa_i = \kappa_i{}' - i\sigma_i/\omega \tag{16}$$

one gets the complex form for the effective field in each grain (capacitor 1 corresponds to {100}ₚc grain family, and capacitor 2 corresponds to {111}ₚc grain

family). Finally, the redistributed field in different grain families is obtained as:

$$E_{100} = \frac{\kappa_{111}}{\nu_{111}\kappa_{100} + \nu_{100}\kappa_{111}} E_{app}$$
$$= \left[ \frac{\tau\kappa'_{111}\omega^2 + \sigma_{111}}{(1 + \omega^2\tau^2)(\nu_{100}\sigma_{111} + \nu_{111}\sigma_{100})} \right.$$
$$\left. - i\frac{\tau\omega\sigma_{111} - \omega\kappa'_{111}}{(1 + \omega^2\tau^2)(\nu_{100}\sigma_{111} + \nu_{111}\sigma_{100})} \right] E_{app}$$
$$= E'_{100} - iE''_{100} \tag{17}$$

$$E_{111} = \frac{\kappa_{100}}{\nu_{111}\kappa_{100} + \nu_{100}\kappa_{111}} E_{app}$$
$$= \left[ \frac{\tau\kappa'_{100}\omega^2 + \sigma_{100}}{(1 + \omega^2\tau^2)(\nu_{100}\sigma_{111} + \nu_{111}\sigma_{100})} \right.$$
$$\left. - i\frac{\tau\omega\sigma_{100} - \omega\kappa'_{100}}{(1 + \omega^2\tau^2)(\nu_{100}\sigma_{111} + \nu_{111}\sigma_{100})} \right] E_{app}$$
$$= E'_{111} - iE''_{111} \tag{18}$$

The subscript represents grain orientation with respect to the external field vector, $\tau = \frac{\nu_{100}\kappa_{111} + \nu_{111}\kappa'_{100}}{\nu_{100}\sigma_{111} + \nu_{111}\sigma_{100}}$ is the relaxation time of the two-grain system, and $E'_{hkl}$ and $E''_{hkl}$ are the real and imaginary components of the effective electric fields, respectively.

## Data availability

The data that support the findings of this study are available from the authors on request.

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

## Acknowledgements

J.D. acknowledges financial support from Australian Research Council Discovery projects DP120103968 and DP130100415. The European Synchrotron Radiation Facility is acknowledged for the provision of experimental beam-time. T.R. would like to acknowledge the Slovenian Research Agency for financial support through the research program P2-0105. Zhen Zhou and Hana Ursic are acknowledged for PFM and conductive AFM data, while Jana Cilensek is acknowledged for AFM sample preparation.

## Author contributions

L.L. and T.R. synthesized the polycrystalline BiFeO$_3$. L.L., J.D. and M.D.M. performed the diffraction experiments. L.L. conducted the data analysis with assistance from J.D., L.L., J.D., D.D. and T.R. all contributed to the interpretation of the data. L.L. wrote the manuscript with input from all authors.

## Additional information

**Competing interests:** J.D. is director of Critus Pty. Ltd. The other authors declare no competing interests.

