## [Peer Review File · Nature Communications]

Reviewers' comments:

Reviewer #1 (Remarks to the Author):

This is an interesting report of results from diffraction investigations of electric-field-induced strains in bulk BiFeO₃. Decoupling of field-induced extrinsic strain (associated with domain-wall movement) and intrinsic strain (associated with unit cell distortions) is noted. In addition, the "leading" and "lagging" of the strain response with respect to field was noted. The authors propose a model for the strain behaviour observed based on conduction along crystallographically defined domain walls.

While it is commendable that the model proposed is capable of reproducing aspects of the electric-field-induced strain behaviour seen, I find it far too speculative to be convincing. It is an hypothesis which has been insufficiently demonstrated in my view. In short, additional independent experimental evidence (not just the strain response) would be necessary for me to be in any way confident that the conducting domain walls are indeed responsible for the unusual field-induced strain behaviour seen.

One possibility might be to tie the measurements made into similar electrical impedance spectroscopy measurements - this would at least verify, or otherwise, that experimentally determined frequency-dependent transport behaviour is consistent with that needed to generate the frequency-dependent strain behaviour found under the assumptions of the model used.

A minor point is that it wasn't clear to me that bipolar electric fields had been used in the study - the data looked to be unipolar. If so, I might also be concerned that strain "imprint" and electric-field-induced strain phase lagging or leading might not be distinguishable. Perhaps this needs more thought.

Reviewer #2 (Remarks to the Author):

The paper reports an interesting observation of different response of the lattice strain and strain due to domain switching in BiFeO₃ ceramic. The authors have carried out a state-of-the art experiment involving high energy x-ray-diffraction in-situ with electric field at different frequencies in combination with macroscopic strain measurements. They show that while the strain due to domain switching lags in phase with respect to the driving field, the lattice strain leads! This is a qualitatively new observation. It appears to provide a microscopic explanation of the macroscopic electromechanical response of BiFeO₃ reported earlier (Adv. Funct. Mater. 25, 2099 (2015)). There is also a good attempt at explaining the phenomenon using a phenomenological model by invoking time dependent electric field distribution by considering different conductivities of the domain walls. The paper is worth publication in Nat. Comm.

I have some points the clarity of which may help the reader.

1. Keeping in view that a similar work on BS-PT (PRB 86, 024104 (2012)) reports that both domain switching and lattice strain decreases with increasing frequency, It would be worthwhile to discuss the difference between the two observations.

2. It is a general observation, emphasized in a recent paper (JAP 120, 154104 (2016)), that the lattice strain follows domain switching. This is common for most piezoceramics. In this context, it would be worthwhile justifying in what sense the authors imply "decoupling" in the title of this paper. Is the difference in the phase angles between the lattice strain and domain switching sufficient to conclude "decoupling"? If not, I may suggest the authors to slightly change the title.

3. Since the experiment has orientation (ψ) dependent data, one is curious to know if the phase angle in the lattice strain and domain switching fraction is angle dependent or is constant. The total lattice strain does sums up all the contribution and hence this information is not available in the presented data.

4. It may be good to write (either in the text or the caption) the field/time at which the $\epsilon_{\text{intrinsic}}$ and ϵ_{macro} was measured in Fig. 3a, b.

Reviewer #3 (Remarks to the Author):

This work presents the domain-wall motion-induced strain and lattice strain may have different trend regarding the contribution to the piezoelectric response, which is very interesting result, providing a clear physical picture that has never been reported previously, which is also critical issue and has been unclear in this field. From this point of view, this work is worthy to be published in NC. However there are several issues that need to be clarified and a major revision is thus required:

1, The authors use the word "decoupling". If the decoupling occurs, it should imply that the domain wall motion strain should not affect the lattice strain's change and vice versa. From my understanding, in this work, the authors have just reported that domain wall motion induced strain and lattice strain have different frequency-dependent trends. They should be still coupled as the lattice strain originates from the ionic displacement that will vary the charge accumulation in domain wall and thus changing domain wall conductivity. They are not independent, i.e. decoupled.

2, The authors M-W analytical model is built based on the conductivities in the range of 10^{-9} - 10^{-10} ($\Omega\cdot\text{cm}$)⁻¹. As the authors comment, the resistivity is reported very diversely. The author should: 1, provide the conductivity of the samples the authors fabricated, and 2, provide the experimental results/evidence using AFM surface potential characterisation to present the feature of conductive domain wall in their samples.

3, The grain and domain size will definitely affect the frequency-dependence. The authors have the samples that are synthesised at different temperatures. It will be nice to have experimental results to see what is trend on frequency-dependent strain change from both aspects.

4, For high frequency characterisation (up to 1000Hz in this work), what are the data deviation (or error bar) from the experiment set up presented in this work.

REVIEWER #1

1. The reviewer wrote: *This is an interesting report of results from diffraction investigations of electric-field-induced strains in bulk BiFeO₃. Decoupling of field-induced extrinsic strain (associated with domain-wall movement) and intrinsic strain (associated with unit cell distortions) is noted. In addition, the "leading" and "lagging" of the strain response with respect to field was noted. The authors propose a model for the strain behaviour observed based on conduction along crystallographically defined domain walls.*

Our reply 1:

We appreciate and thank the reviewer for showing interest in our work and providing a thorough assessment.

2. The reviewer wrote: *While it is commendable that the model proposed is capable of reproducing aspects of the electric-field-induced strain behaviour seen, I find it far too speculative to be convincing. It is an hypothesis which has been insufficiently demonstrated in my view. In short, additional independent experimental evidence (not just the strain response) would be necessary for me to be in any way confident that the conducting domain walls are indeed responsible for the unusual field-induced strain behaviour seen. One possibility might be to tie the measurements made into similar electrical impedance spectroscopy measurements - this would at least verify, or otherwise, that experimentally determined frequency-dependent transport behaviour is consistent with that needed to generate the frequency-dependent strain behaviour found under the assumptions of the model used.*

Our reply 2:

We appreciate the reviewer's comment on the requirement for additional evidence for the model. Firstly, previous independent measurements and analyses on our BiFeO₃ samples are available in the literature; these include dielectric response (Rojac et al., Appl. Phys. Lett. 109, 042904, 2016; not cited in the original manuscript), conductive atomic-force microscopy (*c*-AFM) (Rojac et al., Adv. Funct. Mater. 25, 2099–2108, 2015; originally cited as Ref 7) and atomic-resolution microscopy (Rojac et al., Nat. Mater. 16, 322–327, 2017; originally cited as Ref 27). All these independent and diverse measurements and analyses are consistent with the model of enhanced domain wall conductivity affecting internal field distributions, and thus strain response that was proposed in the current manuscript. Indeed, these prior results provided the motivation for pursuing the use of *in situ* X-ray diffraction (XRD) analysis in the present work, *i.e.*, for the purpose of confirming our hypothesis on the role of conductive domain walls (enhanced electrical conductivity at domain walls) in the unusual converse piezoelectric response with negative phase angle.

We agree with the reviewer that the original manuscript discusses the piezoelectric Maxwell-Wagner model without properly acknowledging the enhanced electrical conductivity at domain walls in polycrystalline BiFeO₃, which is essential to the model, and without providing key data from previous publications that support the Maxwell-Wagner model used in the present manuscript. These issues are now clarified in this

response and the manuscript and supplementary information were modified to include those clarifications.

To correlate the conductive domain walls in our BiFeO₃ samples with the Maxwell-Wagner model proposed in our manuscript, we first revisit and clarify the identification of elevated electrical conductivity at domain walls in our samples. We note that prior studies by the co-authors on the enhanced conductivity of domain walls in polycrystalline BiFeO₃ samples have been cited in the original submission of our manuscript as ref 7 and ref 27 (in the revised manuscript, these refs are now ref 7 and ref 31, respectively). To confirm this essential element of the proposed model, we additionally performed *c*-AFM mapping, combined with piezo-response force microscopy (PFM) imaging on the same sample region, to prove the existence of domain walls with enhanced electrical conductivity in poled samples used for the *in situ* XRD analysis. In addition, we provide electrical impedance and piezoelectric measurements on the samples that were analysed by *in situ* XRD, as proposed by the reviewer, along with supporting data from previously published work, which are additionally elaborated upon to provide answers to the reviewer's queries.

For the same batch of material that is presented in our manuscript, we analysed the local conductivity in different regions in the ceramics using *c*-AFM combined with PFM. PFM was used to visualize domains and domain walls, while *c*-AFM was then used to probe the local electrical current in selected grain regions with domain walls. For direct comparison, same sample region was analysed with PFM and *c*-AFM.

A representative area of the poled BiFeO₃ sample is shown in Figure R 1. The PFM image identifies different domain regions with domain walls separating them (arrows in Figure R 1a) inside grains. The *c*-AFM map shows that these domain walls are indeed associated with an enhanced electrical current (white lines in Figure R 1b marked with arrows, indicating increased current signal at domain walls). Additional evidence is provided by the electric-current profile crossing two domain walls (blue line in Figure R 1b) where two current peaks are clearly observed (Figure R 1c). These results confirm the higher electrical conductivity of domain walls in poled BiFeO₃ relative to that of the domains' interior (*i.e.*, inner regions of the domain away from the domain walls). Since

the orientation of these conductive domain walls is different in different grain families, it is reasonable to assume that the conductivity of individual grains in the direction of the external field axis will vary from grain to grain. This is the essential feature of our Maxwell-Wagner model.

The observed enhancement of domain wall conductivity has been acknowledged in the main manuscript (revised manuscript; page 8, paragraph 2) and has been added in the supplementary information (revised SI; Supplementary Figure 6).

Figure R 1 (a) Out-of-plane (OP) PFM amplitude image, (b) *c*-AFM image of the area indicated with a red box in panel (a), and (c) local electric-current profile across the blue line indicated in panel (b). The analysed sample is poled BiFeO₃ sintered at 780 °C (see methods in the manuscript for details). The white arrows in respective images indicate domain wall positions. The *c*-AFM map and current profile show that the domain walls in the analysed BiFeO₃ sample, unambiguously identified by PFM imaging, are associated with enhanced electrical current signal, confirming their higher electrical conductivity relative to that of the domains' interior. The regular, equally spaced parallel stripes in panel b are related to current signals representing artefacts of current measurements which have no correlation with domain walls, grain boundaries (compare with PFM images) or any other topographical feature. PFM imaging was performed by applying to the tip 6 V of AC voltage, while *c*-AFM imaging was performed by biasing the tip with 22 V DC voltage.

Being essential for our model, PFM and *c*-AFM analyses of conductive domain walls are now clearly acknowledged in the main manuscript (revised manuscript, page 8, paragraph 2) and have been added in supplementary information (revised SI; Supplementary Figure 6). This should clarify the link between the conductive paths (domain walls) in BiFeO₃ and our analytical model. These results are completely consistent with our previous published work, *i.e.*, Rojac et al., Adv. Funct. Mater. 25, 2099–2108, 2015 (originally cited as Ref 7) and Rojac et al., Nat. Mater. 16, 322–327, 2017 (originally cited as Ref 27) in which further details are reported. These papers are now cited as Ref. 7 and Ref. 31,

respectively, in the revised manuscript, and Ref. 7 and Ref. 6 in the revised supplementary information.

After showing evidence of conducting domain walls, we below give evidence why conductivity in domain walls is responsible for the Maxwell-Wagner-type piezoelectric behaviour.

To highlight the consistency of *frequency-dependent charge transport behaviour* and *frequency-dependent strain behaviour*, which the reviewer is proposing, the electrical impedance and equivalent "piezoelectric impedance" data of a sample from the same batch are measured at the same field conditions as the *in situ* XRD experiments (sinusoidal unipolar field, 6 kV/mm amplitude). Figure R 2 shows these data in terms of dielectric permittivity (κ), dielectric loss ($\tan\delta$) and charge-voltage phase angle (δ). The real (κ'), imaginary (κ'') part of the permittivity and dielectric loss ($\tan\delta$) (Figure R 2a-c) show a dispersive trend with decreasing driving frequency. Note that the change of κ'' with frequency (from 6 at 1000 Hz to 7500 at 0.01 Hz) is an order of magnitude larger than that of κ' (from 100 at 1000 Hz to 450 at 0.01 Hz). The total κ'' can be represented by $\kappa'' = \kappa_0 \cdot \kappa_d''(\omega) + \frac{\sigma_0}{\omega}$, where $\kappa_0, \kappa_d''(\omega), \sigma_0$ and ω are vacuum permittivity, frequency-dependent dielectric loss, specific bulk electrical conductivity and angular frequency, respectively (A. K. Jonscher, Dielectric relaxation in solids, Chelsea Dielectrics Press, 1983). Therefore, at low frequencies, κ'' is controlled by the bulk electrical conductivity of the sample ($\frac{\sigma_0}{\omega}$ term) as expected at such large driving fields (6 kV/mm). This is clearly seen by analysing the charge-voltage phase angle. The phase angle, δ , as a function of frequency (Figure R 2d) indeed shows an evolution from $\sim 0^\circ$ (capacitive response) at the high frequency end (1000 Hz) towards 90° (resistive/conductive response) at the low frequency end (0.01 Hz). Typical features of Maxwell-Wagner-like behaviour (step-like dispersion in κ' and local peaks in κ'' and $\tan\delta$) depend on measuring parameters and material properties and may not be visible under all conditions; in this case they are probably masked by the dominating contribution from the bulk electrical conductivity at 6 kV/mm field. Indeed, Maxwell-Wagner-like behavior

clearly emerges in the dielectric response measured at weak fields (0.02 kV/mm), which is shown next.

Figure R 2 Dielectric permittivity measured at the same condition as the *in situ* XRD experiments (unipolar 6 kV/mm field) on a sample from the same batch: (a) real part (κ'), (b) imaginary part (κ'') of the permittivity, (c) tangent of phase angle ($\tan\delta$; dielectric loss), and d) phase angle in degrees (δ) as a function of driving frequency.

Figure R 3 shows the permittivity measured at weak bipolar field (0.02 kV/mm) as a function of both frequency and temperature for a BiFeO₃ sample used for the *in situ* XRD analysis (Rojac et al., Appl. Phys. Lett. 109, 042904, 2016). These data are consistent with predictions of a Maxwell-Wagner-like dielectric response. A step-like increase with frequency in κ' (Figure R 3a, arrows) is obvious for all temperatures. This step increase is associated with a weak and broad peak in κ'' (Figure R 3b), highlighted in the inset of Figure R 3b. The barely visible weak and broad peak in κ'' implies that this peak is probably buried into a high bulk conductivity background. More evident peaks are observed in the tangent of phase angle ($\tan\delta$) (Figure R 3c and d). The temperature dependence of all these features, *i.e.*, κ' step-like behavior and κ'' and $\tan\delta$ peaks, showing a shift to higher frequency with increasing temperature, confirms the thermally activated character of the Maxwell-Wagner mechanism. Finally, after the step-like dispersion, it is noteworthy that κ' shows a further increase with lowering of the frequency, probably indicating Jonscher-type universal relaxation due to hopping

conductivity in the system (Lunkenheimer et al., Phys. Rev. B 70, 172102, 2004). The results are thus also consistent with hopping conductivity due to the presence of electron holes as discussed previously for BiFeO₃ with *p*-type conductive character (Rojac et al., Nat. Mater. 16, 322–327, 2017; originally cited as Ref 27).

Figure R 3 Dielectric permittivity measured at 0.02 kV/mm bipolar cyclic electric field on BiFeO₃ sample: (a) real part (κ'), (b) imaginary part (κ'') (Inset b: an enlarged view of the imaginary part at 144 °C on a logarithmic scale, indicating a weak and broad peak), and (c) tangent of phase angle as a function of frequency. Panel (d) shows an enlarged view of the tangent of phase angle to highlight the phase angle peaks at different temperatures as indicated by the black arrow.

The “piezoelectric impedance” data obtained from the sample used for *in situ* XRD experiments and measured at conditions of the *in situ* XRD analysis (6 kV/mm, unipolar) are provided in Figure R 4. The piezoelectric behaviour is consistent with the Maxwell-Wagner behaviour in permittivity as shown in Figure R 2 and Figure R 3. The overall behavior of the complex piezoelectric coefficient (d' and d'') and phase of the piezoelectric (strain-field) response ($\tan\delta_p$), observable in Figure R 4a-c, suggest Maxwell-Wagner-like behavior of the piezoelectric response (Damjanovic et al., J. Appl. Phys. 90, 5708-5712, 2001; originally cited as Ref 12). In particular, the sequence of decreasing and increasing d' with decreasing frequency (red arrows in Figure R 4a), corresponding to the d'' peaks with minima and maxima (red arrows in Figure R 4b), suggests relaxation and retardation processes, respectively, consistent with the distinct characteristics of piezoelectric Maxwell-Wagner relaxation (Damjanovic et al., J. Appl.

Phys. 90, 5708-5712, 2001; originally cited as Ref 12). Note that piezoelectric relaxation and retardation processes were here for the first time directly identified in XRD analysis via the sign of the piezoelectric phase angle extracted for microscopic strain data (negative – relaxation, positive – retardation; manuscript page 7). Finally, the strong dispersion in d' and d'' at the low frequency end (blue arrows in Figure R 4a,b) is related to irreversible displacements of conducting domain walls, in agreement with previous data on BiFeO₃ ceramics processed using different conditions with respect to the current sample (Rojac et al., Adv. Funct. Mater. 25, 2099–2108, 2015; originally cited as Ref 7).

Figure R 4 Piezoelectric response measured on a sample from the same batch and at the same field (unipolar 6 kV/mm field) as was used by *in situ* XRD experiments: (a) real part (d'), (b) imaginary part (d''), and (c) phase angle ($\tan \delta_p$) of the piezoelectric response. In panel a and b, the red arrows indicate the decreasing and increasing of d' with decreasing frequency and the corresponding d'' peak with minima and maxima, while the blue arrows represent the strong dispersion in d' and d'' at the low frequency end, which is related to irreversible displacements of conducting domain walls (see Rojac et al., Adv. Funct. Mater. 25, 2099–2108, 2015).

A Maxwell-Wagner-like mechanism requires presence of units within the materials with different conductivities. In our model, as presented in the manuscript, those units are grains with different crystallographic orientation in which the conductivity is determined by different orientations of conducting domain walls. Note that in a rhombohedral perovskite structure the bulk conductivity is not expected to be strongly anisotropic so one would not expect significant Maxwell-Wagner behaviour a priori just due to the bulk conductivity and different orientation of the grains (*e.g.*, no evidences exist for negative

piezoelectric phase angle in rhombohedral PZT; see Rojac et al., Phys. Rev. B 93, 014102, 2016). Additional evidence for the contribution of conducting domain walls to the Maxwell-Wagner-like relaxation was presented previously by the co-authors (Rojac et al., Adv. Funct. Mater. 25, 2099–2108, 2015; originally cited as Ref 7; Figure 2). It was shown that the dispersions in both the permittivity and piezoelectric coefficient are strongly field-dependent only at low frequencies, consistent with contribution of depinned domain walls to the electro-mechanical response.

In light of these discussions, we are of the opinion that both the electrical impedance and “piezoelectric impedance” data are consistent with the Maxwell-Wagner behavior seen in the *in situ* XRD data and with the proposed piezoelectric Maxwell-Wagner analytical model. The data obtained using *in situ* X-ray diffraction experiments are the most direct evidence revealing the ferroelastic domain wall contribution to the piezoelectric strain dispersion. The direct observation of conducting domain walls, the dielectric and piezoelectric impedance data, our previously published data (Rojac et al., Adv. Funct. Mater. 25, 2099–2108, 2015; Rojac et al., Appl. Phys. Lett. 109, 042904, 2016; Rojac et al., Nat. Mater. 16, 322–327, 2017) and the current *in situ* XRD data taken together convincingly show that the origin of the Maxwell-Wagner-like frequency dependence is indeed related to domain wall conductivity.

We have revised the manuscript to better emphasize all these points, *i.e.*, the electrical and piezoelectric impedance measurements (revised manuscript; Page 7; paragraph 2). These data have also been included in the supplementary information for clarity (revised SI; Supplementary Figure 3-5). The mentioned prior publication, which was not originally cited (Rojac et al., Appl. Phys. Lett. 109, 042904, 2016), has been added to the reference list in the revised manuscript (revised manuscript; Ref 30).

3. The reviewer wrote: *A minor point is that it wasn't clear to me that bipolar electric fields had been used in the study - the data looked to be unipolar. If so, I might also be concerned that strain "imprint" and electric-field-induced strain phase lagging or leading might not be distinguishable. Perhaps this needs more thought.*

Our reply 3:

The reviewer is correct: the field applied during the *in situ* XRD experiment is unipolar. With unipolar fields, we observed the electric-field-induced macroscopic strain lagging (positive phase) and the lattice strain leading (negative phase) the applied electric field waveform. The negative phase angle (phase leading) was also observed in the direct d_{33} measurements at small unipolar stresses (Rojac et al., J. Appl. Phys. 112, 064114, 2012). The negative phase also appears in the macroscopic converse piezoelectric data with bipolar electric fields when the response is extrapolated to zero field (Rojac et al., Adv. Funct. Mater. 25, 2099–2108, 2015; originally cited as Ref 7).

It can be concluded that the negative phase angle appears at low stresses and electric fields in both direct and converse macroscopic strain response (under both unipolar and bipolar conditions) and can be related to the observed lattice strain measured by XRD. Increasing amplitude of the driving stresses and fields enhance the domain wall displacements, which results in positive phase angle that then dominates the direct and converse macroscopic responses (over the negative phase angle response). In light of all these consistent experimental results, we therefore did not further consider effects of imprint. All different experiments support the presence of a negative phase angle in BiFeO₃ material. Note also that piezoelectric measurements must be made in poled samples, which is by definition an "imprint".

We revised the manuscript (revised manuscript; page 6; paragraph 4) and the publication mentioned above is now included (revised manuscript; ref 24). The revised text now reads “It is worth noting that the negative phase angle (phase leading) was also observed in the direct d_{33} measurements at small unipolar stresses [24] and in the converse d_{33} with bipolar electric fields when the response is extrapolated to zero field [7]. At low stresses and fields, the negative phases of both direct and converse macroscopic strain can be related to the observed phase leading of lattice strain directly measured by *in situ* XRD here. Increasing driving stresses and fields will enhance the domain wall displacements, which results in positive phase angle that then dominates the direct and converse macroscopic responses. All these different and mutually consistent experiments confirm the presence of negative phase angle in BiFeO₃.”

REVIEWER #2

1. The reviewer wrote: *The paper reports an interesting observation of different response of the lattice strain and strain due to domain switching in BiFeO₃ ceramic. The authors have carried out a state-of-the art experiment involving high energy x-ray-diffraction in-situ with electric field at different frequencies in combination with macroscopic strain measurements. They show that while the strain due to domain switching lags in phase with respect to the driving field, the lattice strain leads! This is a qualitatively new observation. It appears to provide a microscopic explanation of the macroscopic electromechanical response of BiFeO₃ reported earlier (Adv. Funct. Mater. 25, 2099 (2015)). There is also a good attempt at explaining the phenomenon using a phenomenological model by invoking time dependent electric field distribution by considering different conductivities of the domain walls. The paper is worth publication in Nat. Comm.*

Our reply 1:

We thank the reviewer for the positive comments.

2. The reviewer wrote: *1. Keeping in view that a similar work on BS-PT (PRB 86, 024104 (2012) reports that both domain switching and lattice strain decreases with increasing frequency, It would be worthwhile to discuss the difference between the two observations.*

Our reply 2:

The paper that the reviewer is referring to is cited as ref 21 in our original manuscript, which is discussed in the introduction.

We first note that in the work by Jones *et al.* (original Ref. 21) the domain switching was showed to increase with decreasing frequency for both tetragonal (Figure 4 in their publication) and monoclinic phases (Figure 5-6 in their publication). The lattice strain for the tetragonal phase shows roughly a constant value in the frequency range measured (*i.e.*, lattice strain is independent on the frequency, Figure 5 in their publication). The authors ascribed this effect to the independence of intrinsic response on frequency. However, the

increase of lattice strain coefficient with decreasing frequency for monoclinic phase (Figure 5 in their publication) was also ascribed to domain switching contribution in their discussion. The complexity may rise from the field-induced process of phase interchange, and we do not argue the validity of the conclusions of that prior publication.

In any case, the frequency dispersions of lattice strains in this previous work by Jones *et al.* and our work are distinct. We argue that this difference is due to the pronounced domain wall conductivity that leads to anisotropy in the conductivity of different grain families in our material. The local conductive paths inside grains, as has been explained in detail in our main manuscript, result in field shielding for lattice strains, especially at lower frequencies with more pronounced domain wall motion and charge migration, and this causes the decrease of lattice strain with decreasing frequency. We agree with the reviewer that the difference is noteworthy and we thus added a comparison between these two cases in the main text for clarification. The new text reads “More specifically, the frequency dispersion of lattice strain (decrease of lattice strain with decreasing frequency) is distinct from previous reports (frequency-independent lattice strain) [22]. Based on the above analytical model, this difference can be explained by domain walls in BiFeO₃ acting as conductive paths within individual grains. Therefore, shielding of the applied external electric field occurs in given grain families, especially at lower frequencies with more pronounced domain wall motion and charge migration, resulting in the observed decrease of lattice strain with decreasing frequency.” (revised manuscript; page12; paragraph 1) .

3. The reviewer wrote: *2. It is a general observation, emphasized in a recent paper (JAP 120, 154104 (2016)), that the lattice strain follows domain switching. This is common for most piezoceramics. In this context, it would be worthwhile justifying in what sense the authors imply “decoupling” in the title of this paper. Is the difference in the phase angles between the lattice strain and domain switching sufficient to conclude “decoupling”? If not, I may suggest the authors to slightly change the title.*

Our reply 3:

The paper that the reviewer is referring to (Khatua et al., J. Appl. Phys. 120, 154104, 2016) is a good example showing correlation of domain switching and strain with field

amplitude. Indeed there is a strong coupling between the two microscopic strain mechanisms, *i.e.*, lattice strain and domain wall motion induced strain, as has also been observed in other ferroelectric/ferroelastic compositions (Pramanick et al., J. Am. Ceram. Soc. 94, 293-309, 2011; originally cited as Ref 19). The additional reference is now also added (revised manuscript; paragraph 3; page 2; ref 20). The revised text now reads “These strain mechanisms in other ferroelectrics, as observed from field-dependent measurement on PZT and $\text{PbTiO}_3\text{-BiScO}_3$, are considered to be interdependent and coupled through intergranular elastic constraints between neighboring grains or within clusters of grains.” Note also that the paper referred to by the referee does not include frequency dependence.

We have used the word “decoupling” in our paper to refer to the variation in the microscopic strain contributions in the *frequency domain*. We agree that this might not be clear for the reader and thus we have re-defined our use of the word "decoupling" in the introduction. The new text in the introduction now reads, “By experimentally separating the lattice strain from the change in non-180° domain texture over the frequency range from 0.01 to 1000 Hz, it is shown that the two strain mechanisms in different grain orientations are decoupled in the frequency domain.” We have also updated the title from “Decoupled domain-wall motion and lattice strain in BiFeO_3 ” to “Frequency dependent decoupling of domain wall motion and lattice strain in BiFeO_3 ”

We have also removed the expression “decoupling” from the abstract and updated it to specifically emphasise that it refers to the variation of microscopic strain mechanisms *with frequency*. Where applicable, we have updated the term “decoupling” to “frequency-dependent decoupling” or “strain decoupling in the frequency domain” in the revised manuscript and the revised SI.

4. The reviewer wrote: 3. *Since the experiment has orientation (ψ) dependent data, one is curious to know if the phase angle in the lattice strain and domain switching fraction is angle dependent or is constant. The total lattice strain does sums up all the contribution and hence this information is not available in the presented data.*

Our reply 4:

We agree with the reviewer that the phase angle from the two-dimensional diffraction data is worth to be clarified. This should give an overview of the phase information of grains in all orientations. The orientation dependence of the phase angle of the microscopic lattice strain is shown in Figure R 5. The domain wall motion phase angle is not shown due to the limited resolution to which it can be fitted in the time domain.

Figure R 5a shows the applied 6 kV/mm unipolar sinusoidal electric field of 1 Hz while Figure R 5b shows the resulting lattice strain response of 200_{pc} reflection at different angles to the applied electric field (0° means the diffraction vector, \mathbf{q} , of $\{200\}_{\text{pc}}$ grains parallel to the field vector \mathbf{E} , while 90° means perpendicular to the electric field). It is obvious from Figure R 5b that at $\sim 52.5^\circ$ the lattice strain magnitude is approximately zero as has been described using a micromechanical model by Hall *et.al.* (ref 22 in our original manuscript), while at lower \mathbf{q} - \mathbf{E} angles (*e.g.* 0°) the lattice strain is positive and at higher \mathbf{q} - \mathbf{E} angles (*e.g.* 90°) the lattice strain is negative. It is worth noting that at all orientations, the peak values of lattice strain appear before the electric field peak (black dashed line), indicating phase leading of lattice strain at all orientations (corresponding to negative strain-field phase angle).

Figure R 5 Orientation-dependent lattice strain phase measured during 6 kV/mm unipolar sinusoidal electric field: (a) Sinusoidal electric field of 1 Hz, (b) 200_{pc} lattice strain at 1Hz field, (c) tangent of the phase angle of the lattice

response at different orientations with respect to the electric field vector, E , at 1 Hz and 100 Hz (Note that the red arrows indicate the sections where the fitting error is greater since the lattice strain is approximately zero as shown in panel b).

Figure R 5c shows the tangent of phase angle of lattice strain at all q - E orientations at two example frequencies, *i.e.*, 1 Hz and 100 Hz. It is clear that the phase angles ($\tan\delta$) are negative (indicating phase leading of lattice strain) at all orientations. Note that the red arrows (Figure R 5c) indicate the sections where the fitting error is greater since the lattice strain is barely observable for these sections as shown in Figure R 5b.

The above discussion shows that the negative phase angle of lattice strain occurs in multiple grain orientations. We have modified the main text in the manuscript (Revised manuscript; page 6; paragraph 4) and have added these results to the supplementary information (revised SI; page 2; Supplementary Figure 2).

5. The reviewer wrote: *4. It may be good to write (either in the text or the caption) the field/time at which the epsilon_intrinsic and epsilon_macro was measured in Fig. 3a, b.*

Our reply 5:

We agree with the reviewer. The measurement conditions were added in the mentioned figure caption (revised manuscript; page 7; Figure 3) and the main text (revised manuscript; page 6; paragraph 1)

REVIEWER #3

1. The reviewer wrote: *This work presents the domain-wall motion-induced strain and lattice strain may have different trend regarding the contribution to the piezoelectric response, which is very interesting result, providing a clear physical picture that has never been reported previously, which is also critical issue and has bene unclear in this field. From this point of view, this work is worthy to be published in NC.*

Our reply 1:

We thank the reviewer for the positive comments and the careful review.

2. The reviewer wrote: *1. The authors use the word “decoupling”. If the decoupling occurs, it should imply that the domain wall motion strain should not affect the lattice strain’s change and vise versa. From my understanding, in this work, the authors have just reported that domain wall motion induced strain and lattice strain have different frequency-dependent trends. They should be still coupled as the lattice strain originates from the ionic displacement that will vary the charge accumulation in domain wall and thus changing domain wall conductivity. They are not independent, i.e. decoupled.*

Our reply 2:

The reviewer is correct. We rethought the terminology of the word “decoupling”. The reviewer wrote “*the authors have just reported that domain wall motion induced strain and lattice strain have different frequency-dependent trends*” which is exactly what we missed to emphasize when using the term “decoupling”. We thus propose to still use the term “decoupling” but only with additional clarification that it is decoupling of strain mechanisms in the *frequency domain*. See also our response No.3 to Reviewer #2.

We believe lattice strain and domain switching strain on the mesoscopic grain-scale level should still be coupled as the reviewer also commented. What our *in situ* experiments and analytical model highlight is that charge migration on domain walls will induce counter-intuitive frequency dispersion of domain wall motion and lattice strain (while the first contribution shows decreasing trend with increasing frequency, the second contribution shows the opposite behaviour). This implies that the domain wall motion and the lattice

strain are *inversely coupled* as the reviewer has pointed out. We therefore changed the title from “Decoupled domain-wall motion and lattice strain in BiFeO₃” to “**Frequency dependent decoupling of domain wall motion and lattice strain in BiFeO₃**”. The term “decoupling of microscopic strains” is now specifically defined as the “frequency-dependent decoupling of microscopic strains” or “decoupling of microscopic strains in the frequency domain” in the revised manuscript and the revised SI.

3. The reviewer wrote: *2, The authors Maxwell-Wagner analytical model is built based on the conductivities In the range of 10^{-9} - 10^{-10} ($\Omega.cm$)-1. As the authors comment, the resistivity is reported very diversely. The author should: 1, provide the conductivity of the samples the authors fabricated, and 2, provide the experimental results/evidence using AFM surface potential characterisation to present the feature of conductive domain wall in their samples.*

Our reply 3:

We agree with the reviewer's suggestion about providing the electrical conductivity data and evidence for enhanced conductivity at domain walls with respect to the domains' interior (*i.e.*, inner regions of the domain away from the domain walls) in our samples.

We measured the specific bulk electrical conductivity of our samples, which were analysed by *in situ* X-ray diffraction (XRD), using two different methods, *i.e.*, a DC method (current-voltage, I-V, measurements) and AC method (low-frequency permittivity measurements).

Figure R 6a shows the current-density–electric-field (j - E) curve and Figure R 6b shows the real part of electrical conductivity (σ') as a function of driving field frequency for BiFeO₃ ceramics (AC driving field 2.5 V mm⁻¹). As we can see from Figure R 6a, the j - E curve shows a linear relationship, *i.e.*, Ohm's law $j = \sigma_0 E$, where the slope represents the specific electrical conductivity σ_0 of the sample. The specific electrical conductivity using this classical I-V method is $\sim 3.5 \times 10^{-9}$ Ohm⁻¹ m⁻¹.

The real part of ac conductivity (σ') in Figure R 6b can be expressed as $\sigma' = \sigma_0 + \omega\kappa_0\kappa_d''(\omega)$, where σ_0 , ω , κ_0 and $\kappa_d''(\omega)$ denote the specific electrical conductivity, angular frequency, vacuum permittivity and frequency-dependent dielectric loss, respectively (A. K. Jonscher, Dielectric relaxation in solids, Chelsea Dielectrics Press, 1983). It is seen in Figure R 6b that at low frequencies, σ' becomes frequency independent, leveling off at values corresponding to σ_0 , *i.e.*, $\sigma' \sim \sigma_0$ (intercept in panel b). The conductivity using this ac impedance method was thus determined as $\sim 4 \times 10^{-9} \text{ Ohm}^{-1} \text{ m}^{-1}$.

Within the limits of experimental methods used, the specific electrical conductivity of BiFeO₃ determined with the two methods is consistent, showing values on the order of $10^{-9} \text{ Ohm}^{-1} \text{ m}^{-1}$. These values were used as a reference in the analytical modelling presented in the main paper (revised manuscript; page 10; paragraph 3) and are now added to the revised supplementary material (revised SI; Supplementary Figure 7).

Figure R 6 (a) Current-density–electric-field (j - E) curve and (b) real part of electrical conductivity (σ') as a function of driving field frequency for BiFeO₃ ceramics (AC driving field 2.5 V mm^{-1}). The j - E curve in panel (a) shows a linear relationship, *i.e.*, Ohm's law $j = \sigma_0 E$, where the slope represents the specific electrical conductivity σ_0 of the sample (given on the graph). The frequency dependent ac conductivity as σ' in panel (b) levels off at the specific electrical conductivity σ_0 of the sample (given on the graph). See main text for more details.

The local conductivity in different regions of the sample used for *in situ* XRD studies as presented in our manuscript were measured using piezoresponse force microscopy (PFM) coupled with conductive atomic force microscopy (*c*-AFM). PFM was used to visualize domains and domain walls, while *c*-AFM was then used to probe the local electrical current in selected grain regions with domain walls.

A representative area of the poled BiFeO₃ sample is shown in Figure R 7 (same as Figure R 1 in the reply to reviewer #1).

The PFM image identifies different domain regions with domain walls separating them (arrows in Figure R 7a). The *c*-AFM map shows that these domain walls are indeed associated with an enhanced electrical current (white lines in Figure R 7b indicating increased current signal at domain walls). Additional evidence is provided by the electric-current profile crossing two domain walls (blue line in Figure R 7b) where two current peaks are clearly observed (Figure R 7c). These results confirm the higher electrical conductivity of domain walls in poled BiFeO₃ relative to that of the domains' interior (*i.e.*, inner regions of the domain away from the domain walls). Since the orientation of these conductive domain walls is different in different grain families, it is reasonable to assume that the conductivity of individual grains in the direction of the external field axis will vary from grain to grain. This is the essential feature of our Maxwell-Wagner model. See also our Reply No. 2 to Reviewer #1.

The observed domain wall conductivity has been acknowledged in the main manuscript (revised manuscript; page 8, paragraph 2) and has been added in the supplementary information (revised SI; Supplementary Figure 6).

Figure R 7 (a) Out-of-plane (OP) PFM amplitude image, (b) *c*-AFM image of the area indicated with a red box in panel (a), and (c) local electric-current profile across the blue line indicated in panel (b). The analysed sample is poled BiFeO₃ sintered at 780 °C (see methods in the manuscript for details). The white arrows in respective images indicate domain wall positions. The *c*-AFM map and current profile show that the domain walls in the analysed BiFeO₃ sample, unambiguously identified by PFM imaging, are associated with enhanced electrical current signal, confirming their higher electrical conductivity relative to that of the domains' interior. The regular parallel stripes in panel b are related to current signals representing artefacts of current measurements which have no correlation with domain walls, grain boundaries (compare with PFM images) or any other topographical feature. PFM imaging

was performed by applying to the tip 6 V of AC voltage, while c-AFM imaging was performed by biasing the tip with 22 V DC voltage.

4. The reviewer wrote: *3, The grain and domain size will definitely affect the frequency-dependence. The authors have the samples that are synthesised at different temperatures. It will be nice to have experimental results to see what is trend on frequency-dependent strain change from both aspects.*

Our reply 4:

We agree with the reviewer that grain and domain size will affect frequency dispersion of the strain response. Defects, grain size and domain size depend on processing conditions and should thus affect internal field redistribution and frequency dependence of strain. We note that the processing temperatures to which the reviewer is referring to and were used here are 745 °C, 760 °C and 780 °C, resulting in a maximum sintering temperature difference of only 35 °C.

Figure R 8a-c shows scanning electron microscopic (SEM) images of the three samples sintered at different temperatures, *i.e.*, 745 °C, 760 °C and 780 °C. The grain size, expressed as the Feret's diameter mean values (Walton et al., Nature 162, 329–330, 1948), analysed for these images (using the UTHSCSA ImageTool Software) are $2.9 \pm 0.9 \mu\text{m}$ (Figure R 8d), $3.3 \pm 1.1 \mu\text{m}$ (Figure R 8e) and $3.8 \pm 1.8 \mu\text{m}$ (Figure R 8f) for the three samples, respectively. However, it is worth noting that there are occasional large grains (*i.e.*, grains with diameter greater than $\sim 7 \mu\text{m}$; see histogram in Figure R 8f) in the sample sintered at 780 °C due to excessive grain growth at this highest sintering temperature. Small variations in the average grain size in our samples (between 2.9 and 3.8 μm) is expected considering the small maximum difference (35°C) in the sintering temperature. We note that the samples were sintered at such temperatures for the exact purpose of varying the grain size only slightly and thus verify reproducibility of XRD measurements for different processing conditions (*i.e.*, sintering temperature). In particular, we were looking at the reproducibility of the unusual increase of lattice strain with increasing frequency in samples sintered at different temperatures but with comparable microstructure (we show subsequently in this response that this is confirmed in all three samples).

Figure R 8 Scanning electron microscopic (SEM) images of thermally etched samples sintered at (a) 745 °C, (b) 760 °C, (c) 780 °C and the grain size distribution analysed from the SEM images for samples sintered at (d) 745 °C, (e) 760 °C and (f) 780 °C. The grain boundaries were drawn on the images for analysis of the grain size.

Figure R 9a-f shows the PFM in-plane amplitude and phase images of the three samples obtained on an area of 5 x 5 μm to inspect the domain structure. We note that the domain structure analysis was performed on poled samples (as such samples were indeed measured by *in situ* XRD) and were analysed by in-plane PFM imaging, rather than out-of-plane. The reason is that in poled samples the spontaneous polarization of the majority of domains is oriented out-of-plane, resulting in small out-of-plane PFM phase contrast (not shown here), making the analysis of domains difficult. The domains shown in Figure R 9 show regular, lamellar-like morphology, typically observed in BiFeO₃ ceramics (Rojac et al., *Adv. Funct. Mater.* 25, 2099–2108, 2015; originally cited as Ref 7). No obvious qualitative difference is observed in the domain structure of the three samples sintered at different temperatures, consistent with their similar grain sizes (Figure R 8).

Figure R 9 (a-c) In-plane (IP) piezoresponse force microscopy (PFM) amplitude and (d-f) phase images of BiFeO_3 samples sintered at 745 °C, 760 °C and 780 °C, respectively. For PFM analysis, the samples were ground with SiC paper and polished with a diamond paste for ~ 2 h. Sample thickness ranged between 0.2 and 0.5 mm.

Figure R 10 a-f shows the frequency dispersion of macroscopic strain (Figure R 10a-c) and microscopic lattice strain and domain texture obtained from *in situ* XRD data (Figure R 10d-f) for the three samples sintered at different temperatures. Note that the domain texture values had larger errors for the sample sintered at 780 °C due to the presence of bigger grains which impact the quality of the diffraction images, thus values are not shown for this sample. Slightly lower temperatures (745 °C and 760 °C) were chosen to reduce the grain size and thus improve sampling statistics of the X-ray diffraction (XRD) measurements. As explained at the beginning of this response, these specific temperatures were chosen to test the reproducibility of XRD measurements in samples with slightly different grain size. As shown in this figure, the frequency-dependent behaviour of samples sintered at 745 °C, 760 °C and 780 °C is qualitatively similar, showing a good reproducibility of the *in situ* XRD experiments. All the data confirm the increasing trend of lattice strain with increasing frequency.

Figure R 10 (a-c) Macroscopic strain and (d-f) microscopic domain texture (blue) and lattice strain (red) obtained from *in situ* XRD data of the BiFeO₃ samples sintered at 745 °C, 760 °C and 780 °C, respectively.

The manuscript has been revised to acknowledge the negligible influence of grain size and domain structure of samples sintered at different temperatures (*i.e.*, 745 °C, 760 °C and 780 °C) on the *in situ* measurements (revised manuscript; page 12, paragraph 3). These data have been added in the supplementary information (revised SI; supplementary Figure 11-13).

5. The reviewer wrote: 4, For high frequency characterisation (up to 1000Hz in this work), what are the data deviation (or error bar) from the experiment set up presented in this work.

Our reply 5.:

The error bars shown in the main manuscript (page 7, Figure 3b) vary from point to point due to the data acquisition process for the macroscopic strain. Depending on the statistics required in the diffraction data, different frequencies were measured stroboscopically over different numbers of cycles. Within these measurements, sampling of the macroscopic strain data also varied in length. In general, those frequencies where macroscopic strain data was available for a larger number of cycles have smaller errors, as the fitting of the amplitude and phase could be performed more accurately. The

modification indicating the related content has been made in the manuscript (revised manuscript; page 14; paragraph 1), reading “To obtain the required diffraction statistics for different frequencies, the measured number of cycles for the above frequencies were 2000, 600, 200, 90, 20, 20, 20, 10, 10, 10, 7, 3, 1 and 1, respectively. In general, frequencies where macroscopic strain data was measured for a larger number of cycles have smaller errors, as the fitting of the amplitude and phase could be performed more accurately.”

REVIEWERS' COMMENTS:

Reviewer #2 (Remarks to the Author):

I believe that the revised paper addresses the concerns of the reviewers by providing additional information (frequency dependent piezoelectric impedance data) for the sake of direct comparison and completeness. The context of the decoupling has also been clarified, and contrast presented with other systems. This version appears appropriate for Nat.Com.

Reviewer #3 (Remarks to the Author):

The authors have provided sufficient additional information, e.g., C-AFM, impedance spectra and the strain evolution at different angular angles in terms of the reviewers' comments. Although the word 'decoupling' still persists questionable, the authors has redefined the frequency-dependent decoupling. I will possibly accept this new definition. The manuscript can be accepted for publication.

Response letter to reviewers for revised manuscript NCOMMS-18-06415 by Lisha Liu, Tadej Rojac, Dragan Damjanovic, Marco Di Michiel and John Daniels

We acknowledge the reviewers for their comments on our revised manuscript submitted to *Nature Communications* entitled “*Frequency-dependent decoupling of domain-wall motion and lattice strain in BiFeO₃*”. Below we provide responses to the reviewers’ comments.

Reviewer #2 (Remarks to the Author): *I believe that the revised paper addresses the concerns of the reviewers by providing additional information (frequency dependent piezoelectric impedance data) for the sake of direct comparison and completeness. The context of the decoupling has also been clarified, and contrast presented with other systems. This version appears appropriate for Nat.Com.*

Our reply to the reviewer:

We thank the reviewer for their positive comments about our revised version.

Reviewer #3 (Remarks to the Author): *The authors have provided sufficient additional information, e.g., C-AFM, impedance spectra and the strain evolution at different angular angles in terms of the reviewers’ comments. Although the word ‘decoupling’ still persists questionable, the authors have redefined the frequency-dependent decoupling. I will possibly accept this new definition. The manuscript can be accepted for publication.*

Our reply to the reviewer:

We first would like to acknowledge the positive comments from the reviewer.

Regarding the use of “decoupling”, we agree with the reviewer that this word is mostly and normally used in describing interruption of elastic grain interactions in polycrystalline piezoelectrics. However, we would like to mention the following:

- 1) The effect is now clearly and precisely defined in the frequency domain by adding “frequency-dependent” or “as a function of frequency” before/after the term “decoupling” in the manuscript wherever applicable. We re-checked all these parts and made sure that the term will not be misunderstood.
- 2) Motivated by the reviewer’s comments, we also considered terms alternative to “decoupling”, however, it was ultimately decided that “decoupling” (in the way we have presented it) is the least likely to be taken out of context.
- 3) Reviewer #2 has suggested they are comfortable with this usage.